# Pulmonary Artery Embolization in the Management of Hemoptysis Related to Lung Tumors

**DOI:** 10.3390/jpm13111597

**Published:** 2023-11-12

**Authors:** Amandine Claudinot, Frédéric Douane, Olivier Morla, Christophe Perret, Marine Neveu, Francine Thouveny, Antoine Bouvier, José Hureaux, Arnaud Le Guen, Jérémy Jouan, Jean-François Heautot, Antoine Larralde, Damiano Cerasuolo, Emmanuel Bergot, Audrey Fohlen, Jean-Pierre Pelage

**Affiliations:** 1Department of Radiology, CHU de Caen, 14000 Caen, France; fohlen-a@chu-caen.fr (A.F.); pelage-jp@chu-caen.fr (J.-P.P.); 2Department of Radiology, CHU de Nantes, 44000 Nantes, France; frederic.douane@chu-nantes.fr (F.D.); olivier.morla@chu-nantes.fr (O.M.); christophe.perret@chu-nantes.fr (C.P.); marine.neveu@chu-nantes.fr (M.N.); 3Department of Radiology, CHU d’Angers, 49933 Angers, France; frthouveny@chu-angers.fr (F.T.); anbouvier@chu-angers.fr (A.B.); johureaux@chu-angers.fr (J.H.); 4Department of Radiology, Centre Hospitalier Bretagne Atlantique, 56000 Vannes, France; arnaud.le_guen@ch-bretagne-atlantique.fr (A.L.G.); jeremy.jouan@ch-bretagne-atlantique.fr (J.J.); 5Department of Radiology, CHU de Rennes, 35000 Rennes, France; jean-francois.heautot@chu-rennes.fr (J.-F.H.); antoine.larralde@chu-rennes.fr (A.L.); 6Department of Biostatistics and Clinical Research, CHU de Caen, 14000 Caen, France; 7Department of Pulmonology and Respiratory Disease, CHU de Caen, 14000 Caen, France; bergot-e@chu-caen.fr; 8Normandy University, UNICAEN, CEA, CNRS, ISTCT-CERVOxy, 14000 Caen, France

**Keywords:** hemoptysis, pulmonary artery, embolization, lung cancer

## Abstract

(1) Background: Bronchial artery embolization has been shown to be effective in the management of neoplastic hemoptysis. However, knowledge of pulmonary artery embolization is lacking. The aim of this study was to evaluate the safety and efficacy of pulmonary artery embolization in patients presenting with hemoptysis related to lung tumors. (2) Methods: This retrospective study reviewed all consecutive patients with cancer and at least one episode of hemoptysis that required pulmonary artery embolization from December 2008 to December 2020. The endpoints of the study were technical success, clinical success, recurrence of hemoptysis and complications. (3) Results: A total of 92 patients were treated with pulmonary artery embolization (63.1 years ± 9.9; 70 men). Most patients had stage III or IV advanced disease. Pulmonary artery embolization was technically successful in 82 (89%) patients and clinically successful in 77 (84%) patients. Recurrence occurred in 49% of patients. Infectious complications occurred in 15 patients (16%). The 30-day mortality rate was 31%. At 3 years, the survival rate was 3.6%. Tumor size, tumor cavitation and necrosis and pulmonary artery pseudoaneurysm were significantly associated with recurrence and higher mortality. (4) Conclusions: Pulmonary artery embolization is an effective treatment to initially control hemoptysis in patients with lung carcinoma, but the recurrence rate remains high and overall survival remains poor.

## 1. Introduction

Hemoptysis may be a symptom of diverse respiratory conditions, and is due to lung cancer in about 20% of cases [1]. The severity of hemoptysis can also vary, ranging from minimal blood-streaked sputum to immediate life-threatening hemorrhage. The effective management of significant hemoptysis includes several interventional procedures and bronchial artery embolization is now the first-line treatment to control bleeding [2]. In less than 10% of patients, hemoptysis originates from the pulmonary artery [3] and is associated with increased mortality [4]. Hemoptysis arising from the pulmonary artery may be traumatic (Swan–Ganz catheter), inflammatory (Behcet’s disease), infectious (tuberculosis with Rasmussen aneurysm, aspergillosis, etc.) or neoplastic [5]. Several reports have underlined the efficacy of pulmonary artery embolization in cases of infectious [6,7,8,9] or inflammatory [10] disease, as well as in cases of iatrogenic pulmonary artery pseudoaneurysms [11]. Data regarding the use of pulmonary artery embolization in the management of hemoptysis related to lung tumors are still very limited [12].

The goal of the present study was to evaluate the safety and efficacy of pulmonary artery embolization in patients presenting with intractable hemoptysis related to lung tumors.

## 2. Materials and Methods

### 2.1. Patients

From December 2008 to December 2020, all lung-cancer-related patients presenting with severe hemoptysis treated by pulmonary artery embolization in conjunction with or after a failed bronchial artery embolization at the five participating centers were retrospectively included. The study was promoted by the Interbreizh Research Foundation. Ethical committee approval was obtained (registration number 2317). Patients treated with pulmonary embolization for other causes (traumatic or infectious pseudoaneurysms, pulmonary arterio-venous malformations, etc.) during the study period were excluded.

Each patient was categorized by age, sex, previous medical history, histologic type of cancer, disease stage, previous therapies (chemotherapy, immunotherapy and radiation therapy), hemostasis abnormality, hemoptysis volume and hemodynamic status. Hemoptysis severity was graded on the basis of the quantity of expectorated blood: <50 mL of hemoptysis was considered as minimal, 50 to 200 mL as medium, and >200 mL in 24 h as massive. All patients were examined with 1 mm collimation computed tomography after the administration of iodinated contrast material. Acquisitions were usually started 45 s after intravenous injection, to obtain maximum opacification of the pulmonary and bronchial arteries at the same time. Characteristics of the tumor (long diameter, location proximal vs. distal, presence of necrosis or cavitation) were recorded. All included patients presented with pulmonary artery lesions considered to represent hemoptysis: proximal pulmonary artery invasion, irregularity of arterial wall or vessel narrowing, pseudoaneurysm and tumoral occlusion. The presence of enlarged bronchial arteries was analyzed. The presence of ground-glass attenuation or alveolar consolidations was also recorded.

### 2.2. Endovascular Management

The initial approach to massive hemoptysis should always begin with airway management and hemodynamic stabilization. Anticoagulant medications should be held for the appropriate period of time and reversal agents employed if necessary. Airway isolation with bronchial blockers and endobronchial use of iced saline and vasoactive agents are among the conservative methods of hemoptysis management. Procedures were performed by 14 interventional radiologists with 2 to 15 years of experience performing embolization (MD, PhD). All interventional procedures were performed under conscious sedation or general anesthesia depending on the hemodynamic status of the patient with continuous monitoring by the intensive care and pulmonology physicians. After the percutaneous introduction of a 6 to 9 Fr vascular sheath in the femoral vein, the pulmonary artery was selectively catheterized using a guiding catheter and different shapes of 5 Fr catheters. In cases of distal vascular involvement, a 2.4–2.8 Fr microcatheter was used to superselectively catheterize the segmental artery causing bleeding. The choice of embolic agents was left at the operators’ discretion according to the location of the anomalies and operator’s habits. The following embolization parameters were recorded: level of occlusion (distal/segmental, lobar or proximal), type of embolization agents, fluoroscopy time and total radiation dose. Particles, gelatin sponge pledgets, acrylic glue, metallic coils, vascular plugs or stent grafts were selected by the interventional radiologist. The major difference between these types of materials is the end result. The stent grafts correct arteries’ abnormalities while preserving their flow. However, stents can only be used for proximal arterial anomalies due to their caliber and for fairly linear arteries because they are more rigid and find it more difficult to take curves. The other different types of materials allow embolization of the target artery. For patients undergoing bronchial embolization, after percutaneous introduction of a 5 Fr sheath in the femoral artery, selective catheterization of the different bronchial arteries was performed with 4 or 5 Fr catheters of different shapes. Target arteries were then superselectively catheterized with a 2.4–2.8 Fr microcatheter. The embolic agent used was microsphere-sized, between 400 to 900 μm, from different vendors. After embolization, all patients were admitted to the intensive care unit.

### 2.3. Analysis of the Outcome

Technical success was defined as the ability to perform super selective catheterization and embolization of target pulmonary artery branches. Clinical success was defined as cessation of bleeding after embolization. Procedure-associated recurrence of hemoptysis and deaths were evaluated as the main outcomes in two separate analyses. Complications were classified according to the Society of Interventional Radiology grading system (minor vs. major, graded from A to F) [13]. Patient survival for both outcomes was assessed at 30 days, 6 months and at the end of the study, using the Kaplan–Meier estimator. *p* values less than 0.05 were considered statistically significant. Follow-up was defined as the time elapsed between inclusion and December 2020 or the loss to follow-up date.

The Cox proportional hazards model for left-truncated and right-censored data was used in the modeling of the time to the recurrence of hemoptysis and death, in both univariate and multivariate analyses. Potential confounding variables, chosen for their clinical relevance, are indicated in the footnotes of the tables. Analyses were performed using R, version 4.1.1 (R Foundation for Statistical Computing, Vienna, Austria), and RStudio, version 1.4.1717 (Integrated Development Environment for R. RStudio, PBC, Boston, MA, USA).

## 3. Results

### 3.1. Patient Population and Analysis of Imaging

A total of 92 patients (70 men, 22 women) with hemoptysis in the context of cancer aged 63.1 ± 9.9 years treated by pulmonary artery embolization were included (Figure 1). A total of 74 (94%) patients were smokers with an average of 45.1 ± 18.6 pack years (median 40.0, min–max 10–120). Primary lung carcinoma was encountered in 80 (93%) patients, whereas lung metastases and undetermined lesions were identified in 6 and 6 patients, respectively. Epidermoid carcinoma and adenocarcinoma were the most common primary tumors identified in 50 (54%) and 21 (22%) patients, respectively (Table 1). Most of the patients had advanced disease at stage III or IV (97%). Cancer treatment included chemotherapy in 51 (55%), radiotherapy in 11 (12%) and surgical resection in 6 (6%) patients, respectively. Eighteen (20%) patients had hemostasis disorders or were treated with anticoagulation or antiplatelet therapies. Massive and medium hemoptysis were reported in 34 (37%) and 36 (39%) patients, respectively. The remaining patients had minimal but recurrent hemoptysis. Sixty-eight (74%) patients had stable hemodynamic condition and 24 (26%) presented with hemodynamic instability.

Eighty percent of tumors were centrally located. The mean diameter was 75.2 ± 26.5 mm with 72% exhibiting necrosis and 47% cavitation. Arterial wall irregularity encountered in 45% of patients was the most frequent abnormality detected at CT. Tumoral occlusion and pseudoaneurysm caused by tumor invasion were found in 28% and 21% of cases, respectively. Ground-glass attenuation was found in 51% of patients. Hypertrophic bronchial arteries were present in 22 (24%) patients. Imaging results are presented in Table 2.

### 3.2. Embolization Procedure

A total of 12 patients underwent pulmonary artery embolization because of persistent or recurrent bleeding after bronchial artery embolization and 38 (41%) had bronchial and pulmonary embolization carried out at the same time. The remaining 42 (46%) patients had pulmonary artery embolization performed first, based on CT findings suggesting pulmonary artery as the source of bleeding.

Embolization was carried out at the lobar level in 34 (37%) patients and more proximally in 31 (34%) patients (Table 2). The material used for the embolization was coils in 35 (39%) patients (Figure 2), a stent-graft in 29 (32%) patients (Figure 3), acrylic glue in 13 (14%) patients (Figure 4), a vascular plug in 12 (13%) patients and gelatin sponge in 1 (2%) patient. In two patients, the embolization material was not mentioned in the radiological report. Among the 90 patients, 15 patients had a secondary embolization device used including vascular plugs in eight cases, metallic coils in four cases and gelatin sponge in two cases. Mean fluoroscopy time was 25.1 ± 12.5 min (median 21.0; min–max 7.0–51.0) and mean radiation dose was 74,277.0 ± 112,855.8 mGy.cm (median 42,548.0; min–max 2109.0–567,047.0).

### 3.3. Clinical Results and Survival

Ten patients experienced hemoptysis during the procedure. Among them, three patients presented massive hemoptysis and two ruptures of pseudoaneurysm were reported. Technical success was reported in 82/92 (89%) patients and primary clinical success was reported in 77/92 (84%).

Mean follow-up after embolization was 5.2 ± 9.4 months (median 1.7; range 0.0–66.8 months).

Three patients had recurrence within 24 h post embolization.

Of the 77 patients with available mid-term follow-up, delayed recurrence of hemoptysis occurred in 38 (49%) patients. Recurrence occurred after a mean of 71.1 ± 164.6 days (median 11.0; min-max 0–730.0). No difference was found in terms of recurrence rate between patients undergoing simultaneous pulmonary and bronchial artery embolization and those treated only with pulmonary artery embolization. Among patients with recurrence, 76% had proximal or lobar involvement. Variables significantly associated with recurrence in univariate analysis are listed in Table 3, notably tumor size (*p* = 0.004), presence of tumor cavitation (*p* = 0.004) or necrosis (*p* = 0.04). The presence of tumor cavitation (hazard ratio (HR) 6.6, 95% confidence interval (CI) 1.5–29.4, *p* = 0.01) and pulmonary artery pseudoaneurysm (HR 86.0, CI 2.9–2538.4, *p* = 0.009) were independently associated with higher risk of recurrence using multi-variate analysis (Table 4). Recurrence occurred in 48% of patients treated with stents, 43% with coils, 23% with glue and 42% with plugs. Multivariate analysis demonstrated higher recurrence in patients treated with covered stents (HR 5.72, CI 1.3–25.6, *p* = 0.012) or gelatin sponge (HR 33.3, CI 1.0–1066.2, *p* = 0.05). The recurrence rate was lower in patients with epidermoid carcinoma using multivariate analysis (HR 0.13, CI 0.02–0.8, *p* = 0.03).

Infectious complications were reported in 15 (16%) patients, including fever and pneumonia.

Survival after embolization at 1, 3, 6 and 12 months was 69% (95% confidence interval (CI): 59–78%) (Figure 5), 38% (95% CI: 28–48%), 27% (95% CI: 17–36%) and 19% (95% CI: 10–27%), respectively. At 3 years, survival was 3.6% (95% CI: 0–8%). Massive hemoptysis, tumor cavitation (Figure 6) or necrosis were associated with increased mortality (Table 5). There was a trend towards decreased survival in patients with arterial wall irregularity. Simultaneous pulmonary artery and bronchial artery embolization was associated with a significantly reduced mortality (Figure 7). No difference was found in terms of survival between proximal and distal embolization. Variables associated with mortality in univariate analysis are listed in Table 6, notably tumor size (*p* = 0.023), presence of tumor cavitation (*p* = 0.003) or necrosis (*p* = 0.02). Using a multi-variate analysis, the presence of tumor cavitation (HR 5.0, CI 1.8–13.43 *p* = 0.001) and pulmonary artery pseudoaneurysm (HR 13.9, CI 2.0–95.4, *p* = 0.007) were independently associated with shorter survival (Table 7). Multivariate analysis demonstrated shorter survival in patients treated with covered stents (HR 6.2, CI 12.1–18.3, *p* < 0.001) or gelatin sponge (HR 21.2, CI 1.3–349.8, *p* = 0.03). Using multivariate analysis, patients with epidermoid carcinoma had better survival (HR 0.34, CI 0.12–1.0, *p* = 0.05).

## 4. Discussion

Life-threatening or recurrent hemoptysis remains an important clinical concern for chest physicians. CT angiography permits the noninvasive, rapid and accurate assessment of the cause and guides subsequent management [14]. In the large majority of cases, the source of hemorrhage is of systemic origin, with bronchial artery embolization used as a first-line effective treatment. Control of hemoptysis can be achieved in 65 to 92% of cases depending on the cause [15]. The main causes of treatment failure are technical challenges and bleeding originating from the pulmonary artery. However, lesions of pulmonary artery branches are not routinely described in radiological reports. Indeed, one study showed that only 46% of pulmonary arteries abnormalities were identified on the initial CT studies [16]. Patients presenting with large tumors and associated pulmonary artery pseudoaneurysms may benefit from an initial pulmonary artery embolization.

It has been estimated that up to 30% of patients with lung cancer will present hemoptysis and, of these, 10% will experience massive hemoptysis [17]. Hemoptysis can also reveal cancer [1]. In patients with tumor-related hemoptysis, overall survival remains low and recurrence is reported in 20 to 30% of cases [17,18]. There is no difference in terms of survival between patients who had previously undergone radiation therapy or not [19].

There are only few studies reporting the use of pulmonary artery embolization and there is no consensus opinion on indications and technique. In previous reports, some authors recommended pulmonary artery embolization only after the failure or recurrence of bronchial artery embolization [2], whereas others recommended pulmonary artery embolization when an abnormality was detected on pulmonary artery at CT [3,9]. Marcelin et al. reported that 7 out of 12 patients treated with pulmonary artery embolization first underwent bronchial artery embolization. They have not reported any case of the simultaneous embolization of both circulations [12]. However, our data suggest that pulmonary and bronchial embolization performed at the same time may be associated with increased survival. Simultaneous catheterization of both circulations was performed in a few patients when the operator judged that there were pulmonary and bronchial arterial abnormalities. Indeed, hilar tumor lesions can cause pulmonary arterial anomalies but are also associated with systemic tumor hypervascularization. In these cases, it seems interesting to perform simultaneous embolization. This situation is not very common in clinical practice. The difficulty lies in carrying out a double arterial and venous femoral approach for this simultaneous treatment, which extends the procedure time. In cases of severe hemoptysis, the most significant abnormality must be treated first.

In addition, for proximal involvement, there is no consensus opinion regarding whether the pathological branches should be occluded with glue or coils, or if the parent pulmonary artery should be spared using a stent graft. Previous reports recommended stent graft placement for proximal pulmonary abnormalities to maintain vessel patency and minimize lung functional loss [12,15]. In our practice, when feasible, ventilation–perfusion lung scintigraphy is carried out prior to proximal embolization to estimate the respective contribution of each lung to the overall function. Nuclear medicine studies are difficult to obtain on an emergency basis and are often not feasible in patients with unstable hemodynamic condition. A higher recurrence and mortality were found in patients embolized with stent grafts. Treatment of proximal lesions may explain these results.

Despite a high technical success rate and initial clinical efficacy associated with embolization, the overall prognosis remains poor. Not surprisingly, in patients with stage III or IV large lung tumors, the 1-year survival rate was 19% in our study. It is well known that pulmonary artery involvement in lung cancer is associated with increased mortality in patients with hemoptysis [4]. The presence of a large or excavated lesion or a pseudoaneurysm seems to be associated with a poor prognosis. In our series, compared to the experience of Marcelin et al. [12], the survival rate was lower (i.e., 38% versus 67% at 3 months) and the recurrence rate was also much higher (i.e., 49% versus 0%). The differences we found may be due to the small number of highly selected patients (19 patients) in the former study, which may have led to an underestimation of overall mortality and recurrence. In addition, there were little descriptive data about the tumor in their study, especially regarding the size of the tumor. Furthermore, the higher number of treatments using stent graft for proximal lesions in our series, 32% versus 16% for Marcelin [12], may have played an important role. The main treatment failures are due to catheterization failures or ruptured arteries during the procedure. Deaths due to massive hemoptysis during the procedure have been reported in our series and also in several studies [12,20]. The rupture of the pulmonary artery might have been caused by increased pressure during superselective injection and also by the repeated friction of the catheter tip on the fragile pulmonary artery. These pathological arteries are very fragile, which requires great caution during treatment. Stent treatment requires greater operator experience to properly calibrate it. Complications associated with pulmonary artery embolization are mainly infectious complications, as reported in 16% of cases in our study. No complication was noted by Marcelin et al. [12].

Our study has some limitations. It was designed as retrospective, one-sample cohort with no control group. The different types of interventions, small population size, large number of years and interventional radiologists for these procedures could be considered other limitations.

## 5. Conclusions

In conclusion, pulmonary artery embolization is an effective treatment to control hemoptysis in patients with lung carcinoma. However, the recurrence rate remains high and overall survival remains poor, with less than 5% of patients still alive at 3 years. Pulmonary arterial embolization is not technically easy, and the indication should be discussed in a pluridisciplinary tumor board. Special attention should be paid to patients with large, excavated or necrotic tumors or when a pseudoaneurysm is visualized, given the high risk of recurrence and early death.

## Figures and Tables

**Figure 1 jpm-13-01597-f001:**
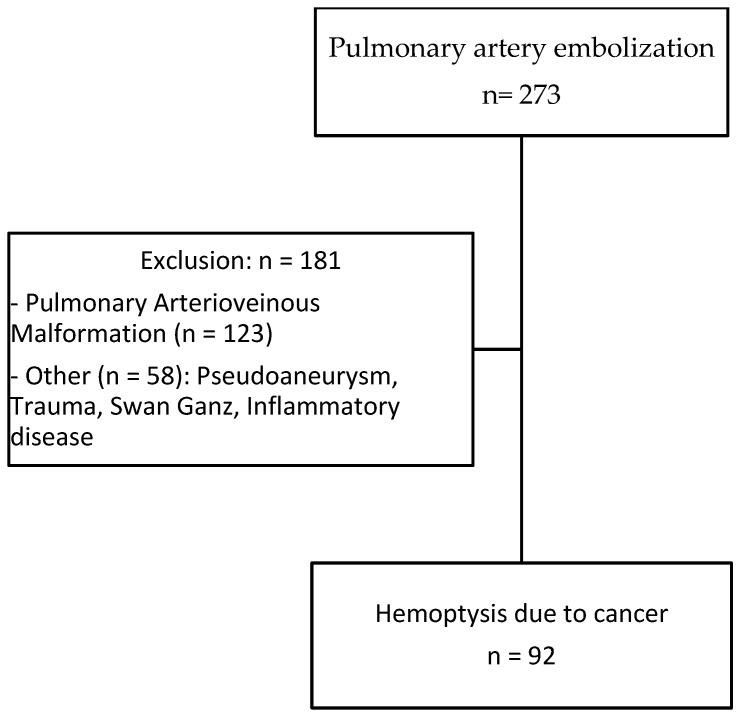
Flow chart.

**Figure 2 jpm-13-01597-f002:**
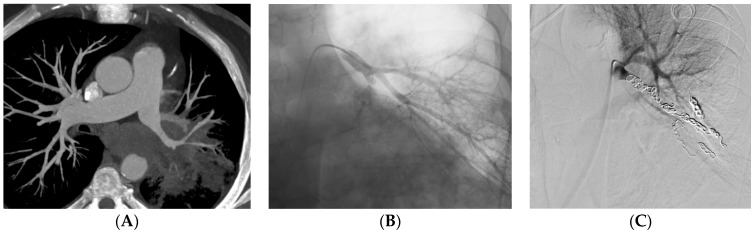
A 62-year-old man with epidermoid carcinoma. He was referred to the RCIU for massive hemoptysis. (**A**) CT before embolization shows irregularity of the left common basal pulmonary artery. (**B**) Basal pulmonary artery angiogram confirms the irregularity. (**C**) Embolization with coils of the left common basal artery angiogram shows complete occlusion.

**Figure 3 jpm-13-01597-f003:**
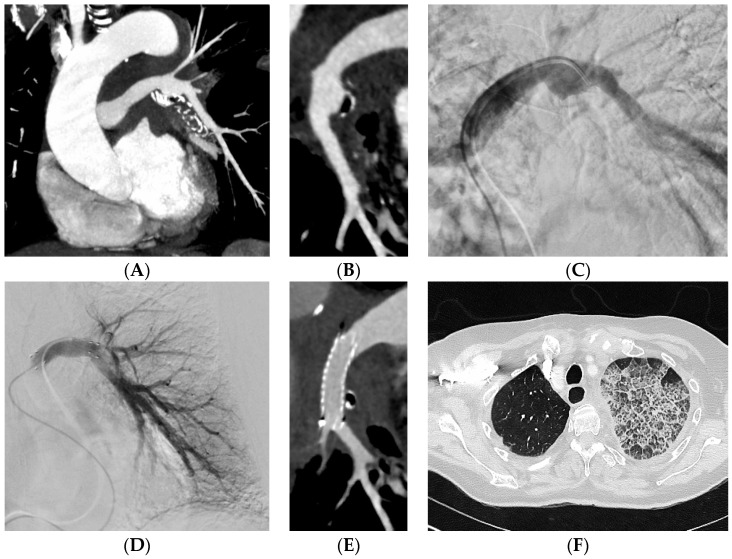
A 79-year-old woman with epidermoid carcinoma. She was referred to the RCIU for minimal hemoptysis. (**A**,**B**) CT before embolization shows pseudoaneurysm of the left proximal pulmonary artery. (**C**) Selective left pulmonary artery angiogram shows the pseudoaneurysm. (**D**) Control angiogram after covered stent placement shows the complete exclusion of the pseudoaneurysm. (**E**) CT after embolization shows the complete exclusion of the pseudoaneurysm. (**F**) CT on parenchymal window shows pneumonia.

**Figure 4 jpm-13-01597-f004:**
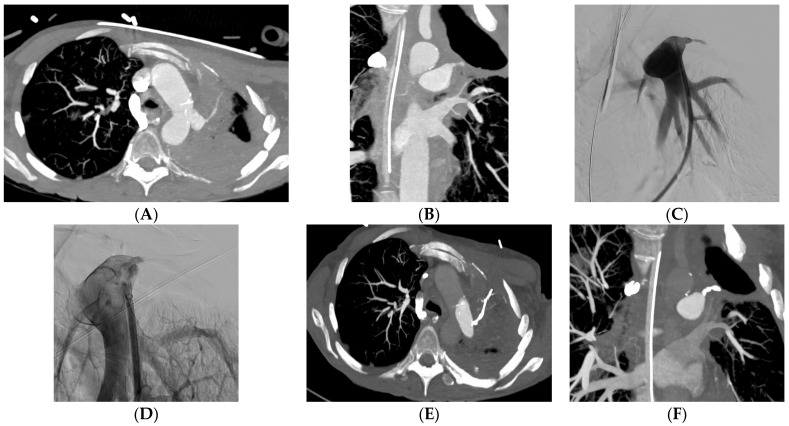
A 65-year-old woman with adenocarcinoma and medium hemoptysis. Axial (**A**) and coronal (**B**) CT shows irregularity of the left anteroapical segmental pulmonary artery wall. (**C**) Superselective left upper artery angiogram confirms the irregularity. (**D**) Control angiogram after embolization with glue shows occlusion of the anteroapical segmental pulmonary artery. (**E**,**F**) CT after embolization shows the occlusion of the left anteroapical segmental pulmonary artery.

**Figure 5 jpm-13-01597-f005:**
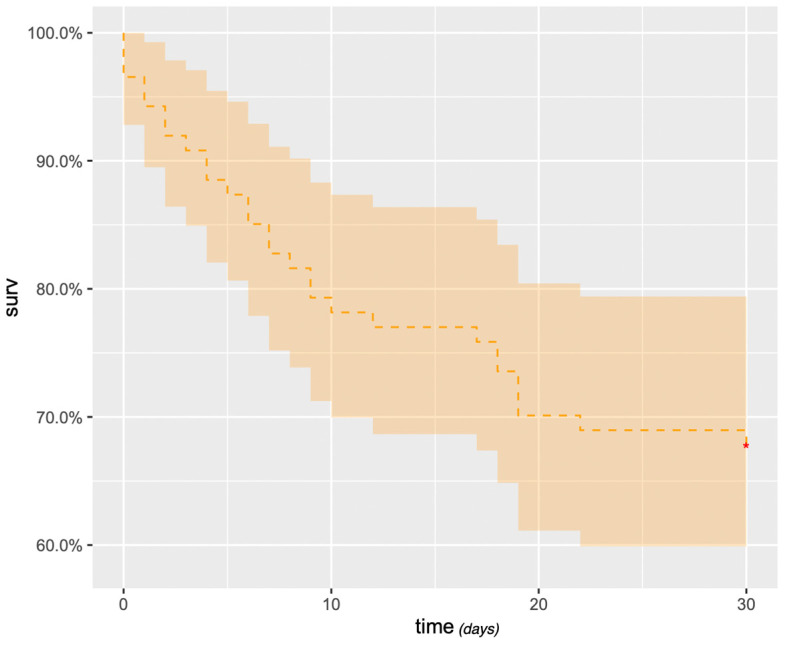
Kaplan–Meier estimate survival rate (days).

**Figure 6 jpm-13-01597-f006:**
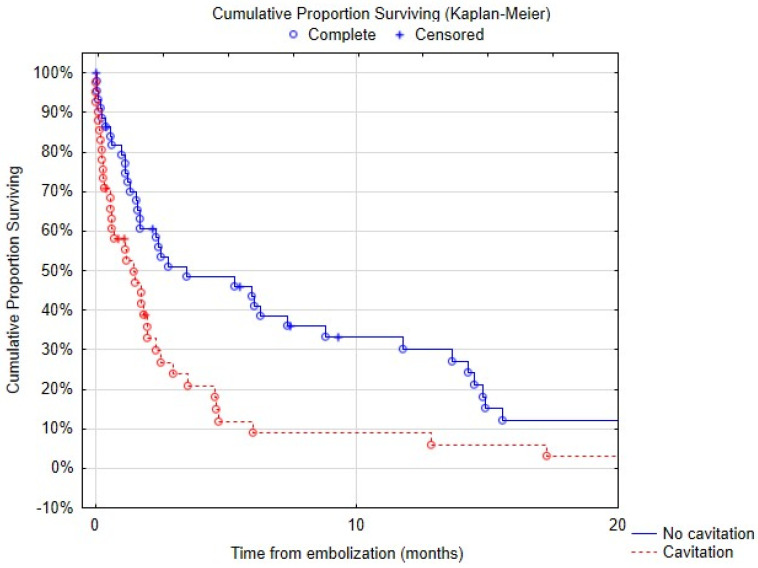
Survival depending on the presence of tumor cavitation.

**Figure 7 jpm-13-01597-f007:**
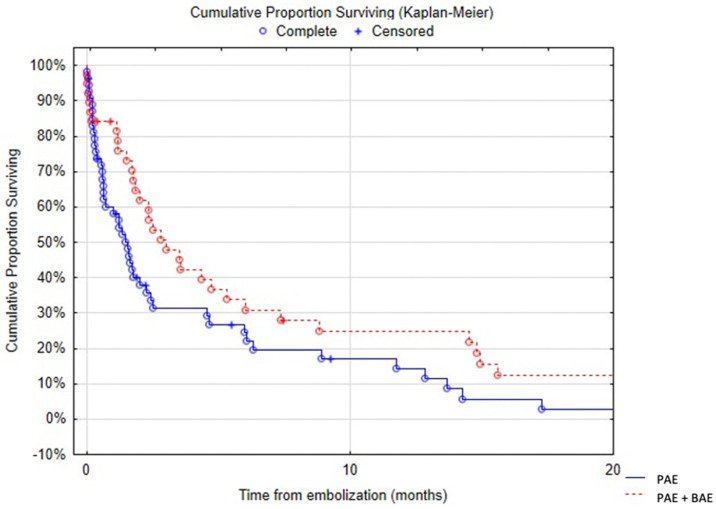
Kaplan-Meier estimate survival rate (days). PAE: pulmonary artery embolization; BAE: bronchial artery embolization.

**Table 1 jpm-13-01597-t001:** Baseline characteristics.

	N = 92
Gender, male/female n	70/22
Mean age (years)	63.1 ± 9.9 (med 63.0, min–max 37.0–85.9)
Histology, n (%) (N = 86)	
Epidermoid	50 (54)
Adenocarcinoma	21 (22)
Lymphoma	2 (2)
Secondary	6 (7)
Undetermined	6 (7)
Other	7 (8)
Staging, n (%) (N = 69)	
I	0 (0)
II	2 (3)
III	22 (32)
IV	45 (65)
Previous treatments, n (%) (N = 92)	
Radiation therapy	11 (12)
Chemotherapy	51 (55)
Immunotherapy	5 (5)
Surgical resection	6 (6)
Altered coagulation n (%) (N = 92)	
Anticoagulation therapy	7 (8)
Antiplatelet therapy	9 (10)
Coagulation disorders	2 (2)
Volume of hemoptysis, n (%) (N = 92)	
Massive	34 (37)
Medium	36 (39)
Minimal	22 (24)
Hemodynamic status, n (%) (N = 92)	
Stable	68 (74)
Instability	24 (26)

**Table 2 jpm-13-01597-t002:** Imaging and embolization findings.

	N = 92
Lung tumor	
Longest diameter (mean SD) mm (73)	75.2 ± 26.5 (70.0, min–max 30.0–162.0)
Tumor necrosis N (%) (N = 87)	63 (72)
Tumor cavitation N (%) (N = 86)	41 (47)
Central location N (%) (N = 91)	73 (80)
Pulmonary artery appearance N (%) (N = 86)	
Arterial wall irregularity	39 (45)
Tumoral occlusion	24 (28)
Pseudoaneurysm	18 (21)
Normal	5 (6)
Parenchymal abnormalities N (%) (N = 37)	
Ground-glass attenuation	19 (51)
Alveolar consolidation	2 (5)
Pulmonary artery: level of embolization N (%) (N = 91)	
Proximal (main arteries)	31 (34)
Lobar	34 (37)
Segmental or more distal	26 (29)
Primary embolization agent N (%) (N = 90)	
Coils	35 (39)
Covered stent	29 (32)
Acrylic glue	13 (14)
Vascular plug	12 (13)
Gelatin sponge	1 (2)

**Table 3 jpm-13-01597-t003:** Univariate analysis of factors associated with recurrence.

	No Recurrence (N = 43)	Recurrence (n = 37)	HR (95%IC)	*p*-Value
Age	62.9 ± 10	62.9 ± 9.7	1 (0.97–1.03)	0.999
Sex (%)				
Female	13 (30.2)	8 (21.6)		
Male	30 (69.8)	29 (78.4)	1.171 (0.53–2.6)	0.697
Histology				
Primary	36 (92.3)	32 (91.4)		
Secondary	3 (7.7)	3 (8.6)	0.7423 (0.22–2.45)	0.625
Histology				
Adenocarcinoma	8 (18.6)	8 (21.6)		
Other	13 (30.2)	8 (21.6)	0.8697 (0.32–2.35)	0.783
Epidermoid	22 (51.2)	21(56.8)	0.7534 (0.32–1.78)	0.519
Volume of hemoptysis				
Minimum	9 (20.9)	9 (21.6)		
Massive	16 (37.2)	17 (45.9)	0.8697 (0.32–2.35)	0.783
Medium	18 (41.9)	12 (32.4)	0.7534 (0.32–1.78)	0.519
Hemodynamic condition				
Instability	12 (27.9)	10 (27.0)		
Stability	31 (72.1)	27 (73.0)	0.6006 (0.28–1.28)	0.189
Altered coagulation				
No risk factor	32 (74.4)	31 (83.8)		
Medication or hemostatic disorders	11 (25.6)	6 (16.2)	0.7475 (0.31–1.8)	0.518
Tumor cavitation				
Absence	22 (55.0)	15 (42.9)		
Presence	18 (45.0)	20 (57.1)	2.7993 (1.37–5.71)	0.004
Tumor necrosis				
Absence	12 (29.3)	11 (31.4)		
Presence	29 (70.7)	24 (68.6)	2.2617 (1.03–4.96)	0.0416
Pulmonary artery				
Normal	4 (10)	1 (2.9)		
Irregularity	19 (47.5)	16 (45.7)	2.7790 (0.35–21.20)	0.324
Occlusion	7 (17.5)	12 (34.3)	2.3885 (0.30–18.70)	0.407
Pseudoaneurysm	10 (25.0)	6 (17.1)	2.8484 (0.34–23.87)	0.335
Tumor size	69.8 ± 24.8	82.0 ± 29.7	1.019584 (1.00–1.03)	0.00351
Proximal embolization				
No	15 (55.6)	8 (32.0)		
Yes	12 (44.4)	17 (68.0)	2.1525 (0.89–5.21)	0.0891
Embolization agents				
Coils	18 (43.9)	14 (37.8)		
Glue	10 (24.4)	3 (8.1)	1.32491 (0.21–2.65)	0.6605
Gelatin sponge	0 (0.0)	1 (2.7)	0.08394 (1.38–102.82)	0.0243
Plug	5 (12.2)	5 (13.5)	1.00006 (0.36–2.79)	0.9999
Covered stent	8 (19.5)	14 (37.8)	0.80773 (0.57–2.69)	0.5902
Immediate bleeding cessation				
No	5 (11.6)	3 (8.1)		
Yes	38 (88.44)	34 (91.9)	0.7617 (0.23–2.51)	0.655

**Table 4 jpm-13-01597-t004:** Multivariate analysis of factors associated with recurrence of bleeding after embolization.

	HR	HR IC: Lower 0.95	HR IC: Upper 0.95	*p*-Value
Age	1.0189	0.9580	1.084	0.5515
Sex				
Female				
Male	5.2860	0.8681	32.188	0.0708
Histology				
Primary				
Secondary	0.9551	0.0182	50.241	0.9819
Histology				
Adenocarcinoma				
Other	0.5557	0.0460	6.708	0.6439
Epidermoid	0.1385	0.0242	0.793	0.0264
Volume of hemoptysis				
Minimal				
Massive	2.0534	0.4870	0.1069	0.6333
Medium	4.5907	0.2178	0.2841	0.2831
Hemodynamic Condition				
Instability				
Stability	0.4662	0.0814	2.671	0.3915
Altered Coagulation				
No risk factor				
Medication or hemostatic disorders	0.4791	0.1068	2.150	0.3368
Tumor Cavitation				
Absence				
Presence	6.5971	1.4792	29.422	0.0134
Tumor Necrosis				
Absence				
Presence	1.1055	0.1844	6.627	0.9126
Pulmonary Artery				
Normal				
Irregularity	13.7288	0.7637	246.792	0.0756
Occlusion	17.2756	0.6156	484.781	0.0940
Pseudoaneurysm	85.9899	2.9129	2538.444	0.0099
Tumor size	1.0213	0.9875	1.056	0.2201
Embolization Agents				
Coils				
Glue	2.4921	0.3385	18.346	0.3700
Gelatin sponge	33.3437	1.0428	1066.195	0.0473
Plug	0.6444	0.0719	5.774	0.6945
Covered stent	5.7254	1.2774	25.662	0.0226
Immediate Bleeding Cessation				
No				
Yes	1.7720	0.1554	20.202	0.6450

**Table 5 jpm-13-01597-t005:** Comparison of survival after embolization using the log-rank test.

Parameters	*p*-Value
Epidermoid vs. adenocarcinoma	0.33
Chemotherapy vs. no chemotherapy	0.63
Heavy smoker (>40 pack years) vs. lighter smoker	0.728
Hilar tumor	0.877
Tumor > 70 mm vs. tumor < 70 mm	0.187
Tumor cavitation	0.006
Tumor necrosis	0.011
Normal vs. abnormal pulmonary artery	0.19
Pulmonary artery abnormality	0.055
Quantification of hemoptysis	0.006
Recurrence of hemoptysis	0.31
Level of embolization	0.202
Proximal vs. distal embolization	0.249
Simultaneous bronchial artery embolization	0.04
Embolization material	0.368
Coils vs. stent	0.309
Coils vs. acrylic glue	0.17
Stent vs. no stent	0.627

**Table 6 jpm-13-01597-t006:** Univariate analysis of factors associated with death vs. survival after embolization.

	Alive (n = 14)	Dead (n = 78)	HR (95% IC)	*p*-Value
Age	67.9 (8.1)	62.3 (10.0)	0.991138 (0.9700972–1.012636)	0.416
Sex (%)				
Female	5 (35.7)	17 (21.8)		
Male	9 (64.3)	61 (78.2)	1.07432 (0.626638–1.841834)	0.794
Histology				
Primary	13 (92.9)	67 (93.1)		
Secondary	1 (7.1)	5 (6.9)	0.6381 (0.2564672–1.587552)	0.334
Histology				
Adenocarcinoma	2(14.3)	19 (24.4)		
Other	2 (14.0)	19 (24.4)	1.004453 (0.5257198–1.919131)	0.989
Epidermoid	10 (71.4)	40 (51.3)	0.727641 (0.4135167–1.280387)	0.2270
Volume of hemoptysis				
Minimum	6 (42.9)	16 (20.5)		
Massive	5 (35.7)	29 (37.2)	2.1008 (1.134935–3.888751)	0.0181
Medium	3 (21.4)	33 (42.3)	2.0151 (1.105948–3.671448)	0.0221
Hemodynamic condition				
Instability	2 (14.3)	22 (28.2)		
Stability	12 (85.7)	56 (71.8)	0.5984 (0.360253–0.9938743)	0.0473
Altered coagulation				
No risk factor	8 (57.1)	65 (83.3)		
Medication or hemostatic disorders	6 (42.9)	13 (16.7)	0.8487 (0.4668211–1.542828)	0.591
Tumor cavitation				
Absence	9 (64.3)	36 (50.0)		
Presence	5 (35.7)	36 (50.0)	2.0597 (1.277593–3.320675)	0.00302
Tumor necrosis				
Absence	6 (42.8)	18 (24.7)		
Presence	8 (57.2)	55 (75.3)	1.9091 (1.113582–3.272895)	0.0187
Pulmonary artery				
Normal	1 (7.2)	4 (5.6)		
Irregularity	4 (28.6)	35 (48.6)	1.3260 (0.4692674–3.746619)	0.594
Occlusion	3 (21.4)	21 (29.2)	0.8468 (0.2862588–2.504777)	0.764
Pseudoaneurysm	6 (42.8)	12 (16.7)	1.3277 (0.4243536–4.154165)	0.626
Tumor size	71.08 (13.32)	75.97 (28.44)	1.01113 (1.0–1.02)	0.023
Proximal embolization				
No	6 (60.0)	20 (40.0)		
Yes	4 (40.0)	30 (60.0)	1.1918 (0.6748679–2.104721)	0.545
Embolization agents				
Coils	9 (64.3)	26 (34.2)		
Glue	2 (14.3)	11 (14.5)	1.7710 (0.8662912–3.620735)	0.1172
Gelatin sponge	0 (0.0)	1 (1.3)	7.1076 (0.9108879–55.459520)	0.0614
Plug	0 (0.0)	12 (15.8)	1.1839 (0.5957885–2.352425)	0.6300
Covered stent	3 (21.4)	26 (34.2)	1.3588 (0.7881746–2.342384)	0.2699
Recurrence of hemoptysis				
No	8 (57.1)	36 (51.4)		
Yes	4 (42.9)	34 (48.6)	0.7623 (0.4703713–1.235272)	0.27
Immediate bleeding cessation				
No	0 (0.0)	8 (10.3)		
Yes	14 (100.0)	70 (89.7)	0.5703 (0.2731161–1.190974)	0.135

**Table 7 jpm-13-01597-t007:** Factor associated with survival using a multivariate analysis (Cox model).

	HR	HR IC: Lower 0.95	HR IC: Upper 0.95	*p*-Value
Age	1.01594	0.97192	1.0620	0.48405
Sex				
Female	1 (Ref)			
Male	1.60732	0.53484	4.8303	0.39794
Histology				
Primary	1 (Ref)			
Secondary	1.32056	0.16292	10.7037	0.79452
Histology				
Adenocarcinoma	1 (Ref)			
Other	0.65678	0.15641	2.7580	0.56581
Epidermoid	0.33961	0.11544	0.9992	0.04982
Volume Of Hemoptysis				
Minimal	1 (Ref)			
Massive	4.49076	0.22268	0.80803	0.08609
Medium	5.44441	0.18367	1.20513	0.02763
Hemodynamic Condition				
Instability	1 (Ref)			
Stability	0.89437	0.32958	2.4271	0.82652
Altered Coagulation				
No risk factor	1 (Ref)			
Medication or hemostatic disorders	0.64646	0.23181	1.8029	0.40447
Tumor Cavitation				
Absence	1 (Ref)			
Presence	4.95697	1.84528	13.3159	0.00150
Tumor Necrosis				
Absence	1 (Ref)			
Presence	0.93443	0.30310	2.8808	0.90602
Pulmonary Artery				
Normal	1 (Ref)			
Irregularity	5.31143	0.98012	28.7834	0.05278
Occlusion	5.15053	0.69212	38.3285	0.10946
Pseudoaneurysm	13.91491	2.03048	95.3590	0.00734
Tumor size	1.02046	0.99964	1.0417	0.05412
Embolization Agents				
Coils	1 (Ref)			
Glue	2.33382	0.72102	7.5541	0.15731
Gelatin sponge	21.24199	1.28991	349.8082	0.03251
Plug	0.91880	0.28363	2.9764	0.88770
Covered stent	6.20609	2.10414	18.3046	0.00094
Immediate Bleeding Cessation				
No	1 (Ref)			
Yes	0.81238	0.22729	2.9036	0.74918
Recurrence of hemoptysis				
No	1 (Ref)			
Yes	0.26032	0.09907	0.6840	0.00633

## Data Availability

The data supporting the findings of this study are available within the article.

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
