# Peer review of "Pulmonary Artery Embolization in the Management of Hemoptysis Related to Lung Tumors"

_jpm, 2023, doi:10.3390/jpm13111597_

Round 1
Reviewer 1 Report
Comments and Suggestions for Authors
This is an excellent manuscript about a retrospective review of the use of pulmonary artery embolization in the management of hemoptysis related to tumors in the lungs. The authors identified the study cohort by retrieving the list of patients who underwent pulmonary artery embolization, and then narrow down to those who underwent the procedure because of hemoptysis due to cancer. An alternative strategy to identify the cohort is to start with the cancer patients with hemoptysis and tumors present in the lungs who underwent arterial embolization (bronchial artery embolization included) for management of hemoptysis. This alternative approach would generate more data and analyses relevant to hemoptysis management.
Author Response
Dear reviewer,
Thank you for your insightful comments. Indeed it would have been interesting to carry out this cohort in this way.
Respectfully
Reviewer 2 Report
Comments and Suggestions for Authors
The article presents a very interesting issue, the one of embolization in order to stop hemoptysis in lung carcinoma, with both advantages and disadvantages. The paper presents aspects regarding the short term and long term outcomes. Due to the scarce information published so far in regard to this issue, we consider that the paper brings a high degree of novelty.
Comments on the Quality of English Language- minor English revisions are needed
Author Response
Dear reviewer,
Thank you for your feedback
Sincerely